# Mesenchymal Stem Cell-Derived Exosomes as a Treatment Option for Osteoarthritis

**DOI:** 10.3390/ijms25179149

**Published:** 2024-08-23

**Authors:** Anupama Vadhan, Tanvi Gupta, Wen-Li Hsu

**Affiliations:** 1National Center for Geriatrics and Welfare Research, National Health Research Institutes, Yunlin 632007, Taiwan; anupama1810@nhri.edu.tw; 2Institute of Clinical Medicine, College of Medicine, National Cheng Kung University, Tainan 701401, Taiwan; tanvigupta0108@gmail.com; 3Regenerative Medicine and Cell Therapy Research Center, Kaohsiung Medical University, Kaohsiung 807378, Taiwan

**Keywords:** osteoarthritis, mesenchymal stem cells, chondrocytes, synovium, exosomes

## Abstract

Osteoarthritis (OA) is a leading cause of pain and disability worldwide in elderly people. There is a critical need to develop novel therapeutic strategies that can effectively manage pain and disability to improve the quality of life for older people. Mesenchymal stem cells (MSCs) have emerged as a promising cell-based therapy for age-related disorders due to their multilineage differentiation and strong paracrine effects. Notably, MSC-derived exosomes (MSC-Exos) have gained significant attention because they can recapitulate MSCs into therapeutic benefits without causing any associated risks compared with direct cell transplantation. These exosomes help in the transport of bioactive molecules such as proteins, lipids, and nucleic acids, which can influence various cellular processes related to tissue repair, regeneration, and immune regulation. In this review, we have provided an overview of MSC-Exos as a considerable treatment option for osteoarthritis. This review will go over the underlying mechanisms by which MSC-Exos may alleviate the pathological hallmarks of OA, such as cartilage degradation, synovial inflammation, and subchondral bone changes. Furthermore, we have summarized the current preclinical evidence and highlighted promising results from in vitro and in vivo studies, as well as progress in clinical trials using MSC-Exos to treat OA.

## 1. Introduction

Osteoarthritis (OA) is a leading cause of disability in adults. In 2020, there were an estimated 595 million people globally with OA, representing 7.6% of the global population [1]. The total number of OA cases increased by 132.2% from 1990 to 2020, and it is expected that, by 2050, there will be a 74.9% increase in knee OA, a 48.6% increase for hand OA, a 78.6% increase for hip OA, and a 95.1% increase for other joint types of OA [1]. OA is the seventh ranked cause of years lived with disability, and the knee is the most common joint affected by OA [2].

The primary symptoms of OA are pain, stiffness, and swelling in the affected joint, resulting in progressive disability [3]. OA has been linked to several systemic factors, including diet, genetics, bone density, and biochemical influences such as obesity and muscle weakness [4]. Aging and gender are also reported risk factors for OA, with some of these factors influencing the overall body metabolism rather than directly affecting the local shape and structure of the joints [5]. OA is a disease that affects the entire joint, with pathological changes occurring in all of the tissues that make up the joint. These changes include the thickening of the subchondral bone, the development of osteophytes, varying degrees of inflammation in the synovium, ligament degradation, meniscal degeneration and inflammation, and hypertrophy of the joint capsule, as well as modifications to the bursa, local fat pads, periarticular muscles, and nerves, leading to pain, swelling, and stiffness [6]. This can impact any joint in the body, with the hips and knees being the most commonly affected [7]. Usually, the wear and tear of cartilage is balanced out by the repair and regeneration of articular cartilage [8]. However, when the body’s regenerative capacity cannot keep up with cartilage damage, it reduces articular cartilage thickness and causes a gradual loss of cartilage, which results in OA [6,8]. Changes in the composition of the articular cartilage promote cartilage degradation through the disruption in the collagen network, altered proteoglycan content, increased catabolic enzyme activity, inflammatory processes, and impaired chondrocyte function [9]. OA is more common with age and is more prevalent in females than males [10].

Chondrocytes are one of the important components of articular cartilage, and their dysregulation is a major issue in OA [11,12]. Chondrocytes play a crucial role in the production, maintenance, and repair of cartilage by striking a balance between the synthesis of extracellular matrix (ECM) components and the secretion of enzymes that can degrade the cartilage, especially matrix metalloproteinase with thrombospondin motifs (ADAMTSs) [13], and are also responsible for ECM destruction [14]. The ECM can be categorized into the pericellular matrix (PCM), territorial matrix (TM), and interterritorial matrix (ITM), each varying in their composition and organization. Chondrocytes are encased by the PCM, and the combination of these two is known as the chondron [15]. The PCM is essential for the metabolic functions and characteristics of chondrocytes, as it contributes to articular cartilage homeostasis and shields chondrocytes from external pressure. The PCM is further surrounded by the TM, and the ITM is the outermost and largest part of the ECM [15]. The PCM consists of aggrecan, hyaluronic acid (HA), glycosaminoglycans (GAGs), and type IV collagen, while the TM and ITM consist of type II, IX, and XI collagens [15]. The breakdown of the ECM in cartilage is involved in OA initiation. In contrast, ECM destruction in cartilage exposes chondrocytes to factors like immune and catabolic mediators, which stimulate chondrocytes to produce proteoglycan aggrecan and type II collagen [11]. During OA progression, chondrocytes become hypertrophic, creating higher levels of enzymes that degrade cartilage, like matrix metalloproteinase-1 (MMP-1), matrix metalloproteinase-3 (MMP-3), matrix metalloproteinase-13 (MMP-13), ADAMTS-4, and ADAMTS-5. These enzymes further degrade type II collagen and aggrecan, which are important for cartilage formation [13,16]. Synovial inflammation can be triggered by various factors like the dysfunction of mitochondria, cytokines, and damage-associated molecular patterns (DAMPs) like HMGB1 (high-mobility group box-1), advanced glycation end-products (AGEs), S100A8, S100A9 proteins released by the damaged cartilage, dysregulated metabolites, and crystals (monosodium urate, hydroxyapatite, and calcium pyrophosphate dihydrate) present in the synovial membrane [17,18]. In joints affected by OA, there is an increased production of pro-inflammatory cytokines like interleukin-1β (IL-1β), interleukin-1 (IL-1), interleukin-6 (IL-6), interleukin-8 (IL-8), and tumor necrosis factor (TNF-α), which damage the cartilage matrix through the interaction with other factors [19,20,21]. On the other hand, anti-inflammatory and anabolic regulators that include interleukin-4 (IL-4), interleukin-10 (IL-10), and insulin-like growth factor-1 (IGF-1), which are produced to maintain joint homeostasis, are hindered during OA progression [17].

Stem cell therapy has become more popular in recent years due to its potential as a regenerative treatment for various conditions, including OA. Stem cells have the unique ability to differentiate into various cell types and can potentially repair damaged tissues and alleviate age-related degeneration [22]. In OA treatment, stem cell therapy has shown promise in preclinical and clinical studies as a means of reducing inflammation and promoting tissue repair [23]. The different types of stem cells include embryonic stem cells (ESCs), tissue-specific progenitor stem cells (TSPSCs), mesenchymal stem cells (MSCs), umbilical cord stem cells (UCSCs), bone marrow stem cells (BMSCs), and induced pluripotent stem cells (iPSCs), which are categorized based on their regenerative applications [22].

Mesenchymal stem cells (MSCs) are used in the majority of current stem cell therapies for OA due to their multilineage differentiation into joint cell types and immunoregulatory function [24,25]. MSCs can be readily reproduced in culture and have a low level of MHC-1 expression, while lacking MHC-II and costimulatory molecules, making them less likely to trigger an immune response during allogeneic transplantation [26]. They can be sourced from adult tissues like bone marrow and adipose tissue, as well as from neonatal tissues such as the umbilical cord, placenta, and amniotic membrane [27,28]. The immunomodulatory effects of MSCs have been demonstrated in multiple research studies, along with their tissue regeneration capacity [29,30,31]. MSCs also control the release of immune-modulating substances like B cells, T cells, natural killer cells, macrophages, and dendritic cells [32], inducing the differentiation of regulatory T cells, which can suppress inflammatory responses and promote tissue repair [33,34,35]. These properties of MSCs make them a promising tool for chronic disease treatment. However, there are still challenges and limitations in the adoption of stem cell therapy in clinical practice [36], while the potential risk of tumor development following stem cell transplantation is a widely debated topic in the literature. Various factors may influence the possibility of tumor formation after MSC transplantation, such as the age of the donor, the host tissue, the growth regulators expressed by the recipient tissue, and the mechanisms that regulate the behavior of the transplanted MSCs at the target site [37]. The prolonged in vitro expansion of MSCs facilitates the cause of genetic instability and chromosomal abnormalities [38]. Therefore, it is important to find therapies where the side effects can be reduced to the greatest extent.

Importantly, the therapeutic benefits of MSCs have been increasingly linked to their secreted extracellular vesicles, particularly exosomes (MSC-Exos) [39,40]. These small-sized membrane-enclosed structures act as signaling platforms, transferring bioactive molecules like proteins, lipids, and genetic materials from parent MSCs to target cells and tissues [41]. Numerous studies have proven the efficacy of MSC-Exos in alleviating OA symptoms and progression in both in vitro and in vivo models [39,40,42].

In this review study, we discuss the pharmacological and non-pharmacological treatments and regenerative medicines which have the potential to treat OA but which also have side effects, which necessitates other approaches for the treatment of OA. To this end, we provide a more detailed understanding of MSC-Exos and their great potential to treat OA patients. We explain the mechanism of action of exosomes, which have anti-inflammatory effects, modulate immune responses, cartilage regeneration, synovial inflammation, and bone remodeling. The preclinical studies and ongoing clinical trials have also shown that MSC-Exos have an effective therapeutic response in the treatment of OA patients with no severe side effects. Therefore, this approach will bring relief to OA patients in the future if it becomes the first-line treatment, but it still it needs to be monitored in large populations to understand its long-term effects.

## 2. Approaches for OA Treatment

OA is a complex and heterogeneous disease, with various subtypes and different underlying causes. This makes it challenging to develop one-size-fits-all treatments that are effective for all patients [43]. The first and most frequent sign of osteoarthritis (OA) progression at the time of clinical presentation is persistent knee pain. Though the exact cause of the pain is unknown, mechanoreceptors (a sensory receptor that responds to mechanical pressure or distortion) in the synovial cavity and subchondral bone, as well as nociceptor fibers (sensory fibers that respond to stimuli), may be involved [3]. Another theory that links pain to bone friction is that pain occurs when the cartilage is unable to keep two bones at normal distances from one another [44]. Osteoarthritis (OA) causes the swelling and enlargement of the bone, which can occasionally be seen in larger joints like the knee as well as smaller joints like the interphalangeal joints [45]. One of the many pathological changes that occur during OA is bone swelling. Among the alterations are chondrocyte damage, soft-tissue edema, blood circulation obstruction, elevated bone density, and the development of cystic lesions [45,46]. All of these pathological alterations combined cause bone remodeling, which can result in several different things, including synovial effusion, capsular thickening, joint subluxation, marginal osteophytosis, and synovial hyperplasia [46]. The articular cartilage surface is frequently the site of the first pathological alterations in osteoarthritis (OA), with fibrillation developing in focal areas under maximum load [46]. Together with topical, oral, and intra-articular medications, a comprehensive plan for managing OA in a specific patient may involve behavioral, psychosocial, educational, and physical interventions [47]. Currently, non-pharmacological and pharmacological treatments are the two approaches that are mainly available for OA treatment [47,48]. The European Alliance of Associations for Rheumatology (EULAR) and the American College of Rheumatology (ACR) have provided detailed interventions for OA management [47,49].

### 2.1. Non-Pharmacological Approaches

Non-pharmacological therapies are thought to be crucial for enhancing function, lowering pain, and enhancing quality of life. According to the EULAR guidelines, for the non-pharmacological management of knee and hip OA, individualized management plans, self-management strategies, tailored exercise programs, and the uses of assistive devices should be taken into consideration [49]. The ACR guidelines have suggested different types of non-pharmacological methods to improve patients’ conditions. These guidelines include weight management for overweight patients, education, and exercise like walking, swimming, tai chi (Chinese mind–body practice), yoga, cognitive behavioral therapy (CBT), and acupuncture to improve joint movement and strengthen joints in OA patients [47,50]. Different physical, psychosocial, and mind–body approaches are suggested to patients according to their needs and the severity of the disease [47,49]. Exercise is strongly advised for all OA patients, but for knee and hip OA, compared to hand OA, there is a lot more evidence supporting its use, and a much wider range of exercise options have been studied [47].

In addition, diet changes also affect OA symptoms [51]. A study reported that, including exercise, the addition of dietary changes significantly leads to changes in body weight, alleviated pain, enhanced overall functional capacity, and decreased inflammatory indicators in the body to a greater extent than exercise alone [52]. Non-pharmacological approaches are usually combined with pharmacological approaches to obtain effective results in OA patients.

### 2.2. Pharmacological Approach

Nonsteroidal anti-inflammatory drugs (NSAIDs) are the initial line of treatment under the pharmacological approach. NSAIDs, such as ibuprofen and naproxen, can help reduce inflammation and alleviate pain associated with OA [53]. Topical formulations of NSAIDs applied directly to the affected joint can provide localized relief with fewer systemic side effects [47,54]. Corticosteroids or hyaluronic acid injections into the affected joint can provide temporary relief from pain and inflammation in patients who do not respond to NSAIDs or are unable to take NSAIDs [55,56,57]. However, using NSAIDs has limitations, as they have well-known side effects. It has been reported that NSAIDs show cardiovascular adverse effects, including myocardial infarction (MI), atrial fibrillation, and thromboembolic events [58]. Furthermore, gastrointestinal, renal, hepatic, and hematological adverse effects have also been reported in patients [59]. These drugs have drawbacks, in that patients may become addicted and dependent on these drugs, especially for opioid-containing NSAIDs [60]. Also, the long-term use of NSAIDs for OA may not provide sustained pain relief and functional improvement, and the benefits may diminish over time [60]. In cases of advanced OA where conservative treatments have failed, joint replacement surgery, such as total knee or hip arthroplasty, may be recommended to alleviate pain and restore function [61]. However, surgery also has many disadvantages. Surgical procedures can lead to complications such as infection, blood clots, and anesthesia-related issues [62]. Some patients even experience an overall reduction in quality of life due to pain or dissatisfaction with the surgical outcomes [63].

The development of disease-modifying OA drugs (DMOADs) has been facilitated by recent advances in our understanding of OA pathology. DMOADs target the specific biological pathways involved in catabolism, rebuilding the anabolism of cartilage, and their main goal is to modify the underlying disease process [64]. DMOADs are typically small molecules of drugs or biologics administered to patients [64]. There have been various DMOADs, like anakinra and lutikizumab, which target inflammatory processes, such as sprifermin, which targets matrix-degrading enzymes, and lorecivivint, which targets the Wnt pathway to decrease cartilage matrix degradation, and many DMOADs are under clinical trials [65,66]. Research is also ongoing for DMOADs targeting cellular senescence and RNA-based therapeutics to target catabolic enzymes, inflammatory pathways, etc. [66]. However, so far, DMOADs have not yet been shown to be fully effective and safe for treating OA [65,67]. The reasons for that failure include inaccurate assumptions about translating findings from animals to humans, adverse side effects, discrepancies between structural symptoms, and inappropriate structural endpoints [68]. Furthermore, the complexity and heterogeneity of OA make it challenging to plan effective clinical trials for DMOADs. Long-term follow-up is also necessary, and there are few reliable biomarkers, like the cartilage oligomeric matrix protein (COMP), MMPs, aggrecan, IL-1β, TNF-α, etc., for the advancement of the disease [69].

### 2.3. Regenerative Medicines

Regenerative medicines using stem cells and their secreted factors to potentially regenerate or repair damaged joint tissues, including cartilage, bone, and other structures, are another emerging option for OA treatment. According to the studies, focal articular cartilage defects have been successfully repaired using cell-based cartilage tissue engineering techniques, though the results have varied [70]. Autologous mesenchymal stem cells (MSCs) are derived from bone and adipose tissue as they can differentiate into bone and cartilage, and are the most commonly used stem cell types for treating knee OA [64]. Though their therapeutic effects are long-lasting and mediated through paracrine mechanisms, there is uncertainty about whether stem cells are the best tool because they tend to disappear quickly from the target tissue [64,71]. Extracellular vesicles (EVs) are tiny particles produced by stem cells that contain cytoplasmic components [72]. EVs play a role in cellular communication and are promising for therapeutic applications due to their high biocompatibility and targeting activity [73]. Stem cell secretomes and EVs have been shown to have anti-catabolic, immunomodulatory, and regenerative qualities in both in vitro and in vivo studies, supporting the translational potential of this regenerative strategy [74]. Since EVs are derived from cells but are not living cells, they are suggested as next-generation biomarkers and a safer substitute for cell-based therapies [64]. Recipient cells can receive nucleic acids, lipids, proteins, and even mitochondria from EVs [75]. EV surface proteins can bind with target cells through the receptor–ligand interaction, leading to the transfer of mRNAs or miRNAs that modulate protein production and gene expression in recipient cells [76]. MSCs and their secretome consist of soluble factors and EVs [77]. The biological activity of EVs has garnered growing attention in recent years, with numerous studies showcasing their potential as a substitute for MSC-based therapy [77]. The most studied class of EVs, exosomes, have a diameter as small as 30–150 nm, suggesting that they can passively diffuse through tissues [78].

MSC-Exos provide a minimally invasive approach for the targeted delivery of cargo molecules to the injury site to achieve therapeutic response in OA treatment [79]. Studies have shown the effectiveness of MSC-Exos in reducing inflammation, promoting tissue regeneration, and alleviating OA symptoms [80]. Because of their unique properties, like their small size, reduced immunogenicity, and cargo for cell-derived components, they are safe and have a promising role in OA treatment. In recent years, various studies have shown that MSC-Exos can effectively promote chondrogenesis and slow down cartilage degradation in OA conditions [81], and there have also have been clinical studies to show that MSC-Exos can be an effective approach for OA treatment [82]. It has been demonstrated that MSC exosomes have strong anti-inflammatory qualities, which can reduce chronic inflammation, stimulate chondrocyte proliferation and differentiation, and inhibit the degradation of extracellular matrix components, a key pathological process in OA [79].

MSC-Exos provide benefits over traditional techniques and stem cell therapy as a whole (Figure 1). MSC-Exos primarily depend on paracrine mechanisms to achieve their therapeutic effects. This means that they secrete different bioactive molecules that can alter the function of target cells and tissues, including growth factors, cytokines, and signaling molecules [83,84]. With this paracrine approach, the dangers of cell-based therapies, like immune rejection, tumor formation, and uncontrolled differentiation, are circumvented [85]. Exosomes can be isolated, characterized, and administered without the need for cell transplantation, making them a more accessible and potentially safer therapeutic option [86]. MSC-Exos have demonstrated a favorable safety profile in preclinical and early clinical studies [83,87]. Moreover, MSC-Exos are relatively stable and can be stored, transported, and administered without the complex logistics associated with cell-based therapies [85]. MSC-Exos can be obtained from various sources, including bone marrow, adipose tissue, and umbilical cord blood [88]. The availability of multiple sources for MSC-Exos production offers a more diverse and accessible pool of therapeutic agents, which can be tailored according to patient needs [83]. The cargo of MSC-Exos is involved in the activity of target cells through various pathways. These exosomes are estimated to carry more than 850 distinct proteins and more than 150 distinct miRNAs [89]. Exosomes are delivered to the recipient cells, and the contents of the exosomes (the miRNAs and proteins) play a role in the regulation and functions of the target cells through multiple signaling pathways [90,91]. As a whole, exosomes represent a promising alternative with the potential for an improved efficacy and safety profile for OA treatment.

## 3. Exosomes, Biogenesis, and Routes of Administration

Exosomes are categorized as a subcategory of extracellular vesicles (EVs) based on their size. There are different types of EVs, including exosomes (30–150 nm in diameter), microvesicles (100–1000 nm in diameter), and vesicular apoptotic bodies (50-5000 nm in diameter), each with distinct characteristics and release mechanisms [85].

Exosomes are derived from the inward budding of the endosomal membrane, microvesicles are released by the ectocytosis of the plasma membrane, and vesicular apoptotic bodies are released during the apoptotic process [92,93]. Exosomes are also known as small extracellular vesicles (sEVs), have a flotation density ranging from 1.1 to 1.18 g/mL, and contain markers such as ALIX, CD81, and TSG101 [94]. Exosomes are secreted by nearly all cells, tissues, and bodily fluids, including plasma, urine, saliva, tears, gastrointestinal secretions, semen, and breast milk [94,95,96]. As shown in Figure 2, exosome biogenesis starts with an early endosome. An early endosome may include a range of components, such as cell signaling receptors, nutrient transporters, ion channels, adhesion molecules, and polarity markers, along with lipid membranes and extracellular fluid, which play a crucial role in cell maintenance and communication between cells [97,98]. Early endosomes then further combine with the endoplasmic reticulum (ER) or trans-Golgi network (TGN), or their constituents, and form late endosomes (LSEs), later known as multivesicular bodies (MVBs) [98,99]. Further, MVBs undergo the formation of intraluminal vesicles (ILVs) by invagination. By accumulating ILVs, MVBs either undergo degradation through interaction with lysosomes and form endolysosomes or transport to the cell membrane to release exosomes with a lipid bilayer by exocytosis [99,100]. Exosomes, regardless of their cellular source, have a lipid bilayer structure that includes a lipid raft. This raft is composed of cholesterol, sphingolipids, ceramide, and phosphoglycerides. Each exosomal particle has been predicted to contain approximately 4,400 distinct proteins [101,102].

All cell types secrete exosomes, are enclosed within a single outer membrane, and contain a variety of substances; therefore, they are used for various purposes like cell-free therapy, drug delivery cargo, and regenerative medicines [103,104,105]. There are no standard methods for exosome isolation so far. The commonly used practices for exosome isolation are ultracentrifugation, polymer precipitation, size exclusion chromatography, ultrafiltration, and immunoaffinity capture [102]. Conventional exosome isolation methods have the following two main limitations: low yield and the impurity of the extracted exosomes [102].

The ultracentrifugation method is based on the differential sedimentation of the exosomes according to their size and density. However, the process can result in significant exosome loss due to incomplete recovery during the centrifugation steps [106]. The yield of exosomes recovered by ultracentrifugation is typically low, ranging between 10% and 40% of the total exosome population in the sample [107]. Precipitation-based techniques involve the use of commercial kits or reagents to precipitate exosomes from the sample. While these methods are relatively simple and do not require specialized equipment, the yield of the extracted exosomes is frequently low, ranging from 30% to 60% of the total exosome population [108]. In terms of impurities, one of the main sources is the presence of other types of EVs, such as microvesicles and apoptotic bodies, which have a similar size and density range as exosomes [109]. Furthermore, the co-isolation of non-vesicular components, such as protein aggregates and lipoproteins, can introduce impurities and reduce the purity and quality of the exosome samples [109].

Usually, intravenous injection and intra-articular delivery are the common methods of drug delivery. Many factors, including the target cells or tissues and the kind of therapeutic agent encapsulated in the exosomes, affect the delivery method. Intra-articular delivery is frequently used for OA treatment, offering a localized approach that allows the exosomes to effectively target specific cells within the affected joint [40,110]. This method concentrates exosomes at the target site, potentially optimizing the treatment efficacy [111]. The second method is systemic administration, which involves the intravenous delivery of exosomes in the body. Exosomes can enter the circulation and travel through the body’s complex pathways to precisely target cells or tissues by injecting them directly into the bloodstream [111]. This approach has been used to treat tendon–bone insertion injuries in vivo [112]. Another approach to delivering the exosomes in OA joints is a combination of exosomes with hydrogels or scaffolds, which are three-dimensional, water-retentive structures and closely resemble the ECM of the cartilage [113]. In the study by Sang et al., they injected chondrocyte-derived-exosome-incorporated hydrogels in a rat OA model, which induced macrophage polarization from M1 to M2 and the protection of cartilage in OA [113]. Hydrogels can serve as a delivery system for exosomes, enabling their sustained and targeted release to affected regions. Exosomes can be incorporated into hydrogel matrices, facilitating the controlled release of bioactive cargo molecules. These methods establish a nurturing environment that fosters cell growth and specialization while leveraging exosomes to influence cellular activities and stimulate tissue repair [114]. In addition to facilitating cell attachment, migration, and growth, scaffolds offer structural support. By combining the regenerative potential of exosomes with the physical properties of the scaffolds, the integration of exosomes into these scaffolds produces a synergistic effect [115]. Hydrogels and scaffolds have been shown to be effective in promoting cartilage repair and regeneration [113,116]. However, in clinical studies, mostly the intra-articular injection method is employed. Two other methods need further investigation to be used clinically. Figure 3 shows the delivery routes of the exosomes for OA treatment currently in use.

## 4. Mechanism of Action of Exosomes

MSC-Exos facilitate chondrogenesis (cartilage formation) in the OA model primarily through (a) the enhancement of chondrocyte (cartilage cells) proliferation, migration, enlargement, and the inhibition of chondrocyte apoptosis; (b) the prevention of the ECM and cartilage degradation; (c) the modulation of immunoprotective or immunodestructive signals. MSC-Exos promote chondrocyte proliferation [117,118], inhibit cell senescence and pro-inflammatory mediators [119,120], protect the cells from oxidative stress [121], and are involved in cartilage repair [122] and regeneration [123]. Exosomal cargo, including proteins, mRNAs, and miRNAs, can activate signaling pathways such as the PI3K/AKT and MAPK/ERK pathways, which are known to promote chondrocyte proliferation [40]. Exosomes carry bioactive molecules such as growth factors and anti-inflammatory cytokines like IL-10 and TGF-β, which suppress the activation and proliferation of pro-inflammatory immune cells such as T cells and macrophages [124]. MSC-Exos possess a beneficial effect on pain relief in OA [117,125] and improve the OA condition by providing important modulatory proteins or non-coding RNA as parental cells [126].

In other words, MSC-Exos promote cartilage health in OA by enhancing the surrounding matrix, supporting chondrocyte survival and function, and modulating immune responses that impact cartilage integrity [127]. Exosomes’ interaction with immune cells, such as T cells and macrophages, modulate the release of cytokines and microRNAs, thereby reducing inflammation and tissue damage (Figure 4).

### 4.1. Exosomes Derived from Various Cell Types and Underlying Mechanism

MSC-Exos can be derived from MSCs derived from different tissues, organs, or cell types [128,129,130,131]. Table 1 shows a list of different types of MSCs from which MSC-Exos can be produced.

#### 4.1.1. Adipose Mesenchymal Stem Cell (AD-MSC)-Derived Exosomes

AD-MSC-derived exosomes (AD-MSC-Exos) containing miR-99b-3p promote extracellular membrane repair in OA by inhibiting metalloproteinase with thrombospondin motifs 4 (ADAMTS4). ADAMTS4, also known as Aggrecanase-1, is a significant enzyme that plays a crucial role in breaking down the cartilage matrix. It can degrade components of the extracellular matrix. In this study, the sources of ADSC were infrapatellar fat and subcutaneous fat [138]. Furthermore, AD-MSC-Exos were reported to downregulate cell senescence, gama-H2AX foci, and reduce the inflammatory mediators IL-6 and prostaglandins E2 in a cell culture model [139]. AD-MSC-Exos (human abdomen-derived adipose tissue) express miR-93-5p, which targets disintegrin and metalloproteinase with thrombospondin motifs 9 (ADAMTS9), hence reducing autophagy and apoptosis, and targeting inflammatory factors like IL-1β, TNF-α, IL-6, and iNOS [140]. ADAMTS9 has been reported to be a biomarker of OA [132]. Exosomes produced from ADSCs (adipose tissues derived by liposuction) have been shown to reduce OA-induced chondrocyte loss and synovial fibrosis by inhibiting the WNT-β-catenin signaling pathway, mainly by acting on miR-376c-3p [141]. Exosomes derived from infrapatellar fat pad (IFP)-MSCs protect articular cartilage through the inhibition of the mTOR mediated by miR-100-5p and binding to the 3’UTR region. In vivo results revealed that IFP-MSC-Exos reduced OA severity, while the in vitro results showed the inhibition of cell apoptosis, the increase in matrix synthesis, and the decrease in catabolic factor expression [142].

#### 4.1.2. Synovial Mesenchymal Stem Cell (SMSC)-Derived Exosomes

The neuropilin-1 (NRP1) protein regulates the bone metabolism and is overexpressed in the cartilage of OA patients [143,144]. Qiu et al. reported that SMSC-derived exosomes (SMSC-Exos) release miR-485-3p, which helps to prevent cartilage damage in osteoarthritis by targeting the NRP1 protein [145]. Fucoidans, a type of sulfated polysaccharide, are commonly present in brown seaweed cell walls and other marine organisms. They are also well known for their immunomodulatory and anti-inflammatory properties [146]. Fucoidan-treated MSC-Exos containing miR-146b-5p [147] have been shown to decrease chondrocyte autophagy and protect ECM degradation, and also suppress inflammatory response by downregulating tumor necrosis factor receptor-associated factor 6 (TRAF6), which plays a role in the synthesis of inflammatory mediators [147]. Bone morphogenetic protein (BMP-7) is a part of the BMP subfamily and has been shown to play a role in the chondrocyte metabolism. It promotes the production, regeneration, and preservation of matrix molecules and has been found to have chondroprotective properties in animal models of osteoarthritis. Exosome release by BMP-7-overexpressed SMSCs has been reported to promote M2 polarization, and thus reduce inflammation [148]. The miR-140-5p-overexpressing human SMSC-Exos have been reported to improve cartilage tissue regeneration and shield the knee from OA in a rat model by inhibiting Ras-related protein A (RaLA), which further leads to SOX9 inhibition, and hence prevents extracellular matrix degradation in articular chondrocytes (ACs) [39]. By increasing proliferation and migration, decreasing apoptosis, and controlling extracellular matrix secretion in chondrocytes, exosomes derived from SMSCs that overexpress miR-155-5p contribute to the prevention of osteoarthritis [149]. The matrilins are a group of four modular, multisubunit adaptor proteins found in the extracellular matrix. Matrilin-1 and 3 are mainly present in skeletal tissues, while matrilin-2 and 4 have a wider distribution across various tissues, including loose connective tissue [150]. Synovial mesenchymal stem cell-derived exosomes carry matrilin-3 (MATN3), which has been reported to reduce cartilage damage and inhibit the breakdown of the ECM by the inhibition of the PI3K/AKT/mTOR pathway [133]. Kartogenin (KGN) is a small compound found to promote the transformation of synovial fluid-derived mesenchymal stem cells (SF-MSCs) into chondrocytes both in vitro and in vivo. However, it has the limitation of poor water solubility, so it forms a precipitate in the cells, resulting in a low efficacy. Adding KGN to SF-MSCs through engineered exosomes causes KGN to be evenly distributed throughout the cytosol, raising its effective concentration within the cell, and significantly stimulating SF-MSC chondrogenesis both in vitro and in vivo [151]. It has been reported that circular RNA delivery with exosomes is also a good strategy to improve the condition in osteoarthritis. Sleep-related circRNA3503-overexpressed extracellular vesicles derived from SMSCs have been shown to have a promising approach to promote chondrocyte regeneration through Wnta/b, cartilage ECM synthesis, and the inhibition of chondrocyte apoptosis [152]. In another study, SMSC-Exos have been shown to express miR-320c, which inhibits disintegrin and metalloproteinase 19 (ADAM19), beta-catenin, and MYC, and hence inhibits Wnt signaling and suppresses ECM degradation and the apoptosis of chondrocytes [153]. MiR-212-5p-overexpressed SMSCs have been shown to treat degenerative chondrocytes in OA. The miR-212-5p targets ETS Transcription Factor 3 (ELF3), which regulates cartilage remodeling and degeneration through SOX9 and MMP13 upregulation [154].

#### 4.1.3. Bone Marrow Mesenchymal Stem Cell (BMSC)-Derived Exosomes

BMSC-Exos have been reported to inhibit pyroptosis (cell death associated with inflammation) and OA improvement through miR-326 in vitro and in vivo. The miR-326 released by BMSCs inhibits the pyroptosis of cartilage and chondrocytes by inhibiting HDAC3 and STAT1//NF-κB p65 [155]. In OA, it has been reported that BMSC-Exos maintain chondrocyte markers (type II collagen and aggrecan) while preventing inflammatory (iNOS) and catabolic (MMP-13 and ADAMTS5) markers in vivo [156]. Furthermore, exosomes derived from BMSCs have been shown to reduce IL-1β-mediated mitochondrial dysfunction, inhibit p38 and ERK, and upregulate AKT in chondrocytes, which, in turn, inhibits their apoptosis [157]. BMSC-Exos treatment has been shown to relieve pain in Lumbar facet joint osteoarthritis (LFJ OA) in a mouse model [158]. BMSC-Exos carrying miR-9-5p inhibit syndecan-1 (SDC1), which has been known to be upregulated in OA [159,160]. BMSC-Exos were also reported to contain miR-125a-5p, which are reported to promote chondrocyte migration and inhibit cartilage degeneration through the inhibition of E2F2 [134]. E2F2 has been reported to be overexpressed in osteoarthritis synovial tissue [161]. BMSCs also contain another microRNA, miR-140-3p, which has been shown to have an anti-inflammatory function under hypoxic conditions and to slow down the pathogenesis of OA [162]. Exosomes derived from bone marrow mesenchymal stem cells possess immunoregulatory properties capable of modulating both innate and adaptive immune reactions, suppressing excessive inflammation and facilitating tissue healing [56]. In a study, it was demonstrated that exosomes originating from BMSCs facilitated the shift of macrophages from the M1 to M2 phenotype, lowered the levels of inflammatory cytokines IL-1β, IL-6, and TNF-α, and enhanced the secretion of the anti-inflammatory cytokine IL-10. Also, the intra-articular injection of BMSC-derived exosomes resulted in decreased inflammation, the mitigation of cartilage damage, and the suppression of osteoarthritis advancement in a rat OA model [110]. Additionally, M1- to M2-type macrophages are polarized by the lncRNA TUC339, expressed in BMSC-Exos, and suppress inflammation while enhancing chondrocyte activity in vitro and in vivo [163]. Exosomes derived from BMSCs have been shown to prevent ROS generation and damage to the mitochondrial membrane, as well as the protein expression of PINK1 and Parkin [164]. TGF-B1-stimulated BMSC-Exos highly expressed miR-135b. Moreover, miR-135b promotes synovial macrophage polarization and reduces cartilage damage [165]. BMSC-Exos have been reported to reverse the harmful effects of AGEs (advanced glycation end-products), which were used to induce cell damage in chondrocytes, promoting autophagy and mitophagy while inhibiting apoptosis and MMP expression. This suggests that BMSC-Exos could be a promising treatment for OA by regulating chondrocyte health and MMP levels through mitophagy [166]. WNT5A has been reported to play a crucial role in the development of articular joints, including cartilage and bone. In the initial phases of cartilage formation, it stimulates the proliferation and reduces the differentiation of chondrocytes [167]. WNT5A can activate MMPs and decrease cartilage formation and cartilage ECM synthesis in the later stages of bone formation and mature chondrocytes [168,169]. This suggests that WNT5 plays different roles the early and late stages of cartilage development. In recent studies, it has been reported that WNT5A plays an important role in OA pathogenesis. WNT5A activates the NF-κB pathway, leading to cartilage deterioration caused by IL-1β. In another study, it has been demonstrated that WNT5A inhibition reduced the IL-1β-induced type II collagen degradation of rat chondrocytes [170]. Exosomes released by BMSC-overexpressing miR-92a-3p targeted WNT5A, improved chondrogenesis, and suppressed cartilage degradation in an OA mice model [167]. MSC-Exos reduced the hypertrophy markers MMP13 and RUNX2, and increased the expression of chondrogenic genes Col2a1 (type II collagen α1) and aggrecan in chondrocytes extracted from OA model mice. The study also demonstrated that exosomal lncRNA-KLF3-AS1 derived from MSCs acts as a competitive endogenous RNA (ceRNA) by sponging miR-206 to facilitate GIT1 (G-protein-coupled receptor kinase interacting protein-1) expression, thereby promoting chondrocyte proliferation and inhibiting apoptosis [171]. GIT1 has been reported to promote chondrocyte proliferation and suppress apoptosis [172,173,174]. Through the PI3K/Akt/mTOR signaling pathway, MSC-Exos-mediated KLF3-AS1 prevents autophagy and the apoptosis of IL-1β-treated chondrocytes. By triggering the PI3K/Akt/mTOR singling pathway through the transcription factor YBX1 (Y-Box binding protein 1), KLF3-AS1 slows the advancement of osteoarthritis [175]. MSC-Exos have also been reported to regulate the chondrocyte glutamine metabolism by downregulating c-MYC, thereby slowing the advancement of OA [176]. MSC-Exos reduce osteoarthritis-related cartilage degradation by blocking the ferroptosis pathway via the GOT1/CCR2/Nrf2/HO-1 signaling pathway [177].

#### 4.1.4. Embryonic Mesenchymal Stem Cell (E-MSC)-Derived and Umbilical Cord Mesenchymal Stem Cell (hUCMSC)-Derived Exosomes

Exosomes from human E-MSCs are reported to balance the cartilage extracellular matrix synthesis and degradation in an OA mice model [135]. Zhang et al. demonstrated that the articular injection of embryonic MSC exosomes in a rat model promoted cartilage repair [40]. Exosomes derived from hUCMSCs have been shown to promote chondrocyte proliferation and migration, and the downregulation of apoptosis in vitro as well as in an in vivo rat model. Furthermore, it has been reported that hUCMSCs have the potential to reverse IL-1β-induced damage to chondrocytes and influence the polarization of macrophages [178]. Similarly, in another study, it was reported that human umbilical cord-derived hUCMSC-Exos have anti-inflammatory properties that promote the chondrogenic gene (Col2a1) and decrease the expression of MMP13 [136]. Exosomes derived from HUCMSC-Exos have higher levels of miR-100-5p, which has been shown to directly target NADPH oxidase 4 (NOX4) to reduce ROS production and cell apoptosis [179].

#### 4.1.5. Dental Pulp Mesenchymal Stem Cell (DP-MSC)-Derived Exosomes

Exosomes produced by dental pulp mesenchymal stem cells (DP-MSCs) have been shown to effectively improve abnormal subchondral bone remodeling, inhibit osteophytes and bone sclerosis, and reduce synovial inflammation and cartilage degradation in vivo [137]. In a study, it was reported that MSC-derived exosomes primed with IL-1β could enhance their anti-inflammatory activity. These exosomes also showed a high expression of miR-147b and the inhibition of the NF-ҝB pathway in a cell line model [180].

## 5. Preclinical Studies: Efficacy and Safety of Mesenchymal Stem Cell-Derived Exosomes in Osteoarthritis Models

The clinical use of MSCs has gained huge interest during the last few years, and they have been thoroughly researched for the treatment of joint damage and OA [181,182,183]. The isolation of MSCs is from adipose tissue, bone marrow, and synovium. The efficacy of MSCs have also been studied by different research groups to regain the loss of function in damaged tissues and to mitigate the symptoms in cartilage damage or OA [184]. It is now known that MSC-Exos protect bone and cartilage from degradation in OA, thereby increasing the expression of chondrocyte markers such as aggrecan and type II collagen, decreasing inflammatory markers such as iNOS, and reducing catabolic markers such as ADAMTS5 and MMP-13, which helps to protect chondrocytes from apoptosis and the blockage of macrophage activation. Also, the MSC-Exos could diminish OA by stimulating chondrocyte proliferation and migration [42,156]. For the treatment of joint diseases, the differentiation of mesenchymal cells into cartilage, bone, and fat tissues, which are responsible for musculoskeletal repair, have the potential for transplantation. As in the case of allogeneic MSCs, the treatment failed to evoke an immune response, and many research groups have found that the transplantation of MSCs in the diseased joints toward the regeneration of joint tissue or functional enhancement led to the engraftment and differentiation of cell types at a very minimal level [185,186]. There are many preclinical studies that have been performed with MSC-Exos for OA treatments, which are mentioned in Table 2. It has been found that the therapeutic role of MSCs is activated by the secretion of trophic factors, thereby improving regeneration with decreased inflammation [187]. Cathepsin K and Wnt inhibitors have been clinically proven to arrest structural progression and inhibit pain in OA. In therapeutic approaches, chondrocytes have been used as an MSC-derived exosome therapy in in vitro studies for the treatment of OA, which are usually taken from tissues such as gingival, synovium, infrapatellar fat pads, and bone marrow [188].

In one study, the authors compared exosomes derived from induced pluripotent stem cell-derived MSCs and exosomes derived from synovial membrane MSCs for the treatment of OA. The authors mentioned a diameter size for both types ranging between 50 and 150 nm, and both types expressed CD9, TSG101, and CD63. The experimental study was performed on a murine OA model by injecting both exosome types, which reduced the symptoms, and a higher therapeutic effect was observed in the induced pluripotent MSCs compared to the synovial membrane MSCs, while both achieved the migration and proliferation of chondrocytes. Due to the ease of access, autologous induced pluripotent MSCs may serve as a new treatment strategy for OA patients [42,189].

An immunocompetent rat model with temporomandibular joint OA was studied by another research team to examine the role of MSC-Exos in nociceptive behavior, inflammatory response, condylar cartilage, and subchondral bone preservation. They uncovered that it led to decreased inflammation, the inhibition of pain, and thus the proliferation and increase in the matrix and the subchondral bone, with an improvement in joint repair and reconstruction. To monitor the cellular activities, the chondrocyte culture can be used for determining the exosome-mediated response of the joint repair, measuring AKT, ERK, AMPK, and adenosine signaling. It was observed that MSC-derived exosomes inhibited by IL-1β led to an increase in s-GAG synthesis, with the reduction in IL-1β-induced nitric oxide and MMP13 as well as other signaling activations. This model, via the use of MSC exosomes, proved successful in the restoration and reconstruction of OA associated with multiple cellular pathways involved in joint repair [190].

Other studies have emphasized chondrogenesis with the role of miRNA, such as in an OA mice model where the MSC-derived exosomes in infrapatellar fat pads helped in the protection of articular cartilage damage and gait abnormalities through miR100-5p-regulated inhibition via the mTOR-autophagy signaling pathway [142]. Therefore, by perf orming arthroscopic operations in clinics, it is easy to obtain human infrapatellar fat pads from OA patients, which could improve the exosome therapy from bench to bedside. MSC-Exos could enhance proliferation and help in inhibiting the apoptosis of chondrocytes through lncRNA-KLF3-AS1/miR-206/GIT1 in OA [171]. SMSC-Exos with miR-320c overexpression could increase chondrocytes via ADAM19 [153]. MSC-Exos with miR-92a-3p overexpression increased chondrogenesis through the inhibition of cartilage damage by Wnt5a signaling [167]. Some of the research groups modified the exosomes before the systemic administration, such as in case of a rat model with OA, where the transfection of synovial MSCs with miR-140-5p overexpression led to an increase in cartilage tissue regeneration [39]. The exosomal-derived miR-135b led to the enhancement of chondrocytes through the regulation of Sp-1 via the stimulation of MSCs through TGF-β1 [191].

Studies have also shown the anti-inflammatory effects and immune modulation involved in OA treatment, such as in vitro, where MSC-Exos inhibited glutamine, glutamine metabolic proteins, the GSH/GSSG ratio, and inflammatory factors, thereby, in the case of in vivo studies, showing an improvement in tissue inflammation, exercise ability, and chondrocyte function, leading to the mitigation of OA progression [176]. Until now, there has not been a single drug which may be beneficial for all OA patients, but there is a chance that OA drugs, DMOADs, will become the next-generation treatment for OA. If both MSC-Exos and DMOADs are combined for OA treatment, they will be able to target matrix-degrading enzymes, inflammatory cytokines, and the Wnt pathway. Also, the DMOAD development involved with miRNA and cellular senescence will be useful for modifying MSC-derived exosomes to improve the treatment for OA patients [65].

**Table 2 ijms-25-09149-t002:** Preclinical studies based on stem cell-derived exosomes for OA treatment.

Serial No.	Source of Exosomes	In Vitro/In Vivo Study	Role in Therapeutic Approach	References
1.	hBMSCs	In vitro	Chondrocyte proliferation, the downregulation of MMP-13, and the upregulation of SOX9	[192]
2.	AD-MSCs	In vitro	Decreased levels of inflammatory cytokines, including IL-6, TNF-α, NO, and PGE2, as well as MMP-13 expression, with increased levels of anti-inflammatory cytokines, including IL-10 and type II collagen expression	[120]
3.	AD-MSCs	In vitro	The downregulation of senescence-associated β-galactosidase and the accumulation of γ-H2AX, with decreased levels of the inflammatory cytokines IL-6 and PGE2	[139]
4.	SMMSCs and induced pluripotent mesenchymal stem cells (iMSCs)	In vivo (in mice)	Chondrocyte migration and proliferation in both exosomes, but showing a higher therapeutic effect in iMSCs	[42]
5.	MSCs	In vivo (in mice)	Cartilage proliferation and chondrogenic differentiation, and the inhibition of WNT5A expression	[167]
6.	BM-MSCs	In vivo (in mice)	Chondroprotection and the inducement of aggrecan and type II collagen expression, with the inhibition of ADAMTS5, MMP-13, and iNOS	[156]
7.	ESC-MSCs	In vivo (in mice)	Attenuated cartilage damage and increased type II collagen through the decrease in ADAMTS5 expression in the presence of IL-1β	[135]
8.	MSCs	In vivo (in rat)	Increased s-GAG synthesis via IL-1β and inhibited IL-1β-induced MMP-13 and nitric oxide production	[190]
9.	MSCs	In vivo (in rat)	A low dose of MSCs promoted the activation of signaling pathways and chondrogenesis	[193]
10.	BM-MSCs	In vivo (in rat)	The upregulation of COL2A1 and downregulation of MMP-13 in cartilage, and also the upregulation of CGRP and iNOS in dorsal root ganglion	[117]
11.	UC-MSCs	In vivo (in rat)	The downregulation of MMP-13, disintegrin, ADAMTS5, IL-1β, and TNF-α, and the upregulation of type II collagen, ki67, and the IL-1 receptor antagonist	[194]
12.	AD-MSCs	In vivo (in rat)	The downregulation of MMP-13 and collagen X, and the upregulation of collagen type II	[195]
13.	BM-MSCs	In vivo (in rat)	The promotion of KDM6A expression and the activation of SOX9	[196]
14.	BM-MSCs	In vivo (in rat)	The inhibition of M1 and the promotion of M2, with decreased expression levels of IL-6, TNF-α, and IL-1β, and increased expression levels of IL-10	[110]
15.	hPMSCs	In vivo (in rat)	Cartilage repair and regeneration	[197]

## 6. Clinical Trials: Evaluating the Therapeutic Efficacy of MSC-Exos in OA Patients

MSC-Exos have been studied at the preclinical level to determine the chondrocyte differentiation and the signaling pathways for OA treatment via intra-articular injections. In terms of safety and reliability, the intra-articular injection of MSC-Exos has proven to be beneficial in the treatment of the knee in OA phase I and II clinical trials for over 5 years. This was successfully achieved, which led to the easing of pain and improved joint and cartilage repair [185,198]. This approach gained more potential at the translational level, but more studies with adequate safety and efficacy have been under trial or are developing at the clinical level so to benefit most OA patients. Some of the ongoing or completed clinical trials for OA treatments are mentioned in Table 3.

Vega et al., 2015, performed a study on 30 human participants, which were divided into two groups that received both allogeneic BM-MSC and HA (as a control) injections in their knee joints. In both groups, this led to minimal adverse events causing discomfort, inflammation, swelling, and pain in the initial week [182]. Gupta et al., 2016, performed injections of both allogeneic and pooled BM-MSCs (Stempeucel^®^) in the knee joints of participants with different dosages of about 25, 50, 75, and 150 million cells, followed by the administration of 10mg/mL of hyaluronic acid (HA). This study showed the best results with 25 million cells, which showed a 64.8% decrease in the Western Ontario and McMaster Osteoarthritis index (WOMAC) score compared to 50 million cells or the placebo, with decreases of 14.4% and 49.3%, respectively. These results showed signs of improvement irrespective of the fact that they were statistically non-significant [199]. Shapiro et al., 2017, stated that bone marrow aspirate concentrate (BMAC) intra-articular injections were safe and viable for OA treatment, whereas it was figured out that BMAC and saline-treated contralateral knees showed similar results [200]. Also, another study confirmed no significant changes in the treatment and saline groups in terms of cartilage regeneration and pain relief [201]. Hernigou et al., 2021, enrolled 60 human participants with the same OA grade in both knees, which were subjected to treatment with BMAC injections with the same MSC concentration in one knee joint as well as in the subchondral bone of the contralateral knee, which surprisingly showed higher scores in patient-reported outcome measures (PROMs) and magnetic resonance imaging (MRI) with the subchondral bone injection [202].

AD-MSCs with about 100 million cells are also suggested to be safe for intra-articular injection, as it reduces pain, and examinations from arthroscopy, MRI, and histological studies showed a decreased size at the damaged site and cartilage regeneration on the subchondral bone [203]. Freitag et al., 2019, conducted a study to determine the efficacy of AD-MSCs with 100 million cells in one- or two-dose regimens by intra-articular injection compared with the conventional treatment, which showed that the MSC therapy was better than the conventional treatments, thereby causing pain relief with improved function [204]. Garza et al., 2020, involved 37 human participants for a treatment with the administration of stromal vascular fractions (SVFs) with AD-MSCs, which confirmed that it is safe and improved the therapeutic efficacy against OA, also providing pain relief, but no significant changes were observed in the cartilage thickness [205].

An allogeneic UB-MSC and HA hydrogel was administered in seven human participants via intra-articular injection in phase I/II clinical trials, which resulted in no adverse effects and improved PROM scores after 6 months. The arthroscopic and histological studies showed cartilage repair after 1 year of treatment and even after 7 years of treatment, which confirms the safety and efficacy of the cartilage regeneration [206]. Soltani et al., 2019, conducted a clinical study with hPMSCs, which showed improvement after 2 months and increased cartilage thickness after 6 months [207].

Akgun et al., 2015, conducted a study where SM-MSCs were administered by matrix-assisted autologous mesenchymal stem cell implantation (MAMI), which was compared to autologous cultured chondrocytes on the porcine collagen membrane (MACI). This study resulted in a higher performance with the MAMI without any adverse effects, thereby improving the PROM scores as well as the cartilage regeneration. Moreover, the SM-MSCs were more chondrogenic than the BM-MSCs [208].

## 7. Challenges and Future Directions

Despite the advancements made, there are still numerous challenges that need to be addressed. One of the key issues is the ability to achieve the large-scale production of exosomes for clinical applications. The large-scale and consistent production of therapeutic-grade exosomes is essential for clinical translation but can be technically challenging to maintain its functional integrity to reach the therapeutic goal [85]. Developing robust and validated manufacturing processes is crucial to ensure the safety and efficacy of exosome-based therapies [209]. Stem cells can release exosomes with varying compositions and functionalities, which can lead to inconsistent therapeutic outcomes. When employed in regenerative medicine or the treatment of disease, this variation in exosome composition may result in uneven therapeutic outcomes. For example, exosomes produced from mesenchymal stem cells (MSCs) can display varying immunomodulatory characteristics based on the source (adipose tissue and bone marrow), as well as the culture conditions. Exosome cargo can also be further influenced by variables like the donor’s age and health as well as the existence of inflammatory signals. Thus, these discrepancies may influence their effectiveness in facilitating tissue restoration, regulating immune reactions, or administering medicinal substances, potentially resulting in unanticipated clinical outcomes [210,211]. The cargo and function of exosomes can be influenced by the physiochemical stimuli provided through the pre-treatment process. However, the MSC exosomes are in a state of stress due to the pre-treatment conditions and other factors influencing their production yield. It is very important to understand whether engineered exosomes derived under the influence of various factors and stress environments have any severe side effects and can reach the therapeutic goal [212]. The standardization of exosome isolation and characterization is crucial to ensure constant quality and efficacy. This involves creating standardized techniques for separating exosomes from different sources, like bone marrow- or adipose tissue-derived mesenchymal stem cells. To guarantee reproducibility, the exosomes’ size, concentration, and cargo—such as proteins, lipids, and RNAs—should all be evaluated following predetermined protocols. Assessments of variables that can change exosome properties, such as the donor’s age, health, and the existence of inflammatory signals, should also be included in the standardization process [213]. Another challenge is to ensure the targeted delivery of exosomes to the affected joint, and their subsequent internalization by the target cells remains a challenge. Many factors, such as the target tissue’s biological milieu and the surface markers on the exosomes, can affect their innate homing abilities. The exosome surfaces must be modified with particular ligands or antibodies to enable precise targeting. These ligands or antibodies must be able to recognize and bind to receptors on target cells to increase the uptake. Furthermore, the distribution and retention of exosomes can be impacted by the joint microenvironment, which is frequently marked by inflammation and changed permeability, making their delivery more difficult. Barriers like cell membrane characteristics and extracellular matrix components can also impede the internalization process itself [214,215]. The source of exosomes and its microenvironment are very important factors to be used as treatment options in patients so to avoid any progression of diseases. Strategies to improve the targeting and exosome retention in the joint are needed, such as the use of specific ligands or modifications [216,217]. The engineered exosomes face challenges in reaching the targeted efficiency in the drug loading, as it becomes difficult due to inadequate space, and with exogenous drugs penetrating inside of the exosomes which previously contained cargo from their parent cells. It is necessary to optimize the conditions and find new approaches to achieve high efficiency in drug loading, as exogenous approaches are required to load therapeutic drugs [218]. Exosome-based therapies face regulatory challenges due to the complex and dynamic nature of the products, which can impact their clinical development and approval [219]. A thorough evaluation of the safety, immunogenicity, and potential adverse effects of exosome-based therapies is necessary [220,221]. The precise mechanisms by which stem cell-derived exosomes exert their therapeutic effects in OA are not yet fully understood. Improved understanding of the underlying mechanisms can help optimize the formulation and application of exosome-based therapies.

One of the major therapeutic concerns is the donor exosomes. If they are obtained from patients that are associated with many diseases, they might have the chance of enhancing the pathology of the diseases. If the donor exosomes are associated with metabolic disorders, they might reduce the quantity as well as the quality of the exosomes, as well as their therapeutic effect, which hinders the biogenesis and biological activity. The source of the exosomes and their microenvironment are very important factors to be used as treatment options in patients to avoid any progression of diseases. The exosomes obtained from the synovial fibroblasts from the joints of OA patients might have a chance for disease progression by promoting inflammation through the activation of inflammatory cells and cartilage degeneration. It has been found that exosomes obtained from the synovial fluid of OA patients stimulate the production of pro-inflammatory cytokines, chemokines, and metalloproteinases from the M1 type [222]. Another study showed that exosomes derived from the synovial fibroblasts of OA patients have tumor necrosis factor-α (TNF-α), which promotes inflammation [223]. The gene expression of chondrocytes could also be negatively affected due to the exosomes derived from the synovial fluid of OA patients. An increase in the expression of catabolic and pro-inflammatory genes was reported, while a decrease in anabolic genes was observed when treated with articular chondrocytes with exosomes derived from OA patients [224]. Exosomes are also involved in cross-talk between macrophages and chondrocytes, as the chondrocytes help in the stimulation of IL-1β through macrophages [225].

MSC-Exos have emerged as a promising therapeutic avenue for the treatment of preclinical arthritis models [188]. Continuing research is needed to develop scalable and reproducible methods for isolating and purifying exosomes from stem cells. Additionally, the detailed characterization of exosome cargo and their specific mechanisms of action in OA need to be elucidated. Strategies to enhance the targeting and uptake of exosomes by joint tissues affected in OA, such as the synovium and cartilage, could improve the therapeutic efficacy. Combining mesenchymal stem cell-derived exosomes with other interventions like anti-inflammatory drugs or other regenerative approaches may produce synergistic effects in managing OA. Developing patient-specific exosome therapies by using the recipient’s stem cells or targeting the unique disease profile of individual patients could lead to more effective treatments.

Genetic or biochemical engineering of mesenchymal stem cells to enrich exosomes with specific therapeutic molecules could enhance their reparative and anti-inflammatory properties [226,227]. The use of exosomes derived from allogeneic, off-the-shelf stem cell sources could provide a more accessible and scalable option compared to autologous stem cell-derived exosomes. As the field matures, there is a growing opportunity to conduct well-designed clinical trials to evaluate the safety and efficacy of stem cell-derived exosome therapies for different arthritis subtypes. Furthermore, integrating exosome therapies with other emerging technologies like 3D bioprinting, gene editing, and tissue engineering could create innovative treatment solutions for OA [228]. The continued advancement of stem cell-derived exosome research holds great potential for developing new and effective therapies to address the significant unmet needs in the management of various OA conditions.

## 8. Conclusions

Exosomes derived from MSCs have been shown to possess anti-inflammatory and chondroprotective properties. These exosomes can inhibit pro-inflammatory cytokine production while promoting anti-inflammatory factor expression. Additionally, exosomes can suppress the activity of matrix metalloproteinases (MMPs), which are responsible for the degradation of the cartilage extracellular matrix.

Furthermore, exosomes can stimulate the regenerative capacity of cartilage by enhancing the proliferation and differentiation of chondrocytes, as well as promoting the synthesis of extracellular matrix components such as collagen and proteoglycans. This ability to modulate the cellular and molecular processes involved in cartilage homeostasis and repair make exosomes a promising therapeutic approach for OA.

Preclinical studies in animal models of OA have demonstrated the therapeutic potential of exosomes. The intra-articular administration of exosomes derived from various cell sources, including MSCs, chondrocytes, and synovial cells, has been shown to alleviate pain, reduce inflammation, and improve joint function in these models.

Looking towards the future, the development of exosome-based therapies for OA holds a great promise. Strategies include using engineered exosomes with enhanced therapeutic properties, combining exosomes with other regenerative approaches, and exploring targeted delivery methods to optimize their efficacy. Additionally, the identification of specific exosomal cargo components responsible for the observed beneficial effects may lead to the development of more tailored and personalized OA treatments.

In a nutshell, the potential of exosomes in the future of OA therapy is reliable. As a natural, cell-derived delivery system, exosomes offer a unique opportunity to modulate the complex pathological mechanisms underlying OA and promote cartilage regeneration. Continued research and clinical translation of exosome-based therapies may pave the way for more effective and innovative treatments for this debilitating joint disease.

## Figures and Tables

**Figure 1 ijms-25-09149-f001:**
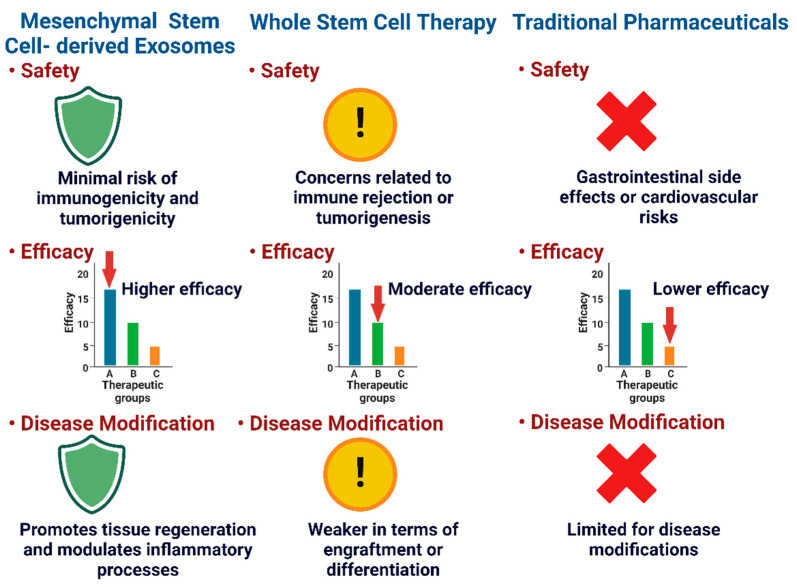
Comparison between MSC-Exos and other therapeutic approaches for osteoarthritis. In safety, the green shield represents higher safety for stem cell-derived exosomes, the yellow circle represents caution for safety in whole-stem-cell therapy, and the red cross represents various safety concerns in traditional pharmaceuticals. In efficacy, the bar graphs show the highest efficacy in stem cell-derived exosomes, moderate efficacy in the whole-stem-cell therapy, and the lowest efficacy in traditional pharmaceuticals. In disease modifications, the stem cell-derived exosomes have the ability to promote regeneration and decrease inflammation, the whole-stem-cell therapy has low chances for engraftment or differentiation, and traditional pharmaceuticals fail to provide any disease modifications. The figure was created with Bio Render (Toronto, ON, Canada).

**Figure 2 ijms-25-09149-f002:**
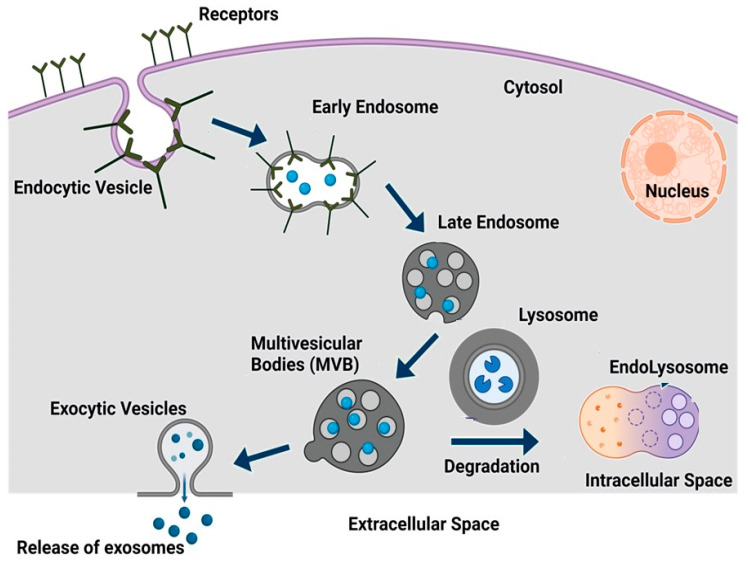
Biogenesis of exosomes. Endocytic vesicles are produced through budding from the plasma membrane to produce early endosomes. Early endosomes further form late endosomes, followed by the formation of multivesicular bodies (MVBs). Some MVBs degrade into lysosomes and form endolysosomes for digestion within the intracellular space. When MVBs and the plasma membranes fuse, exosomes are released into the extracellular space. The figure was created with Bio Render (Toronto, ON, Canada).

**Figure 3 ijms-25-09149-f003:**
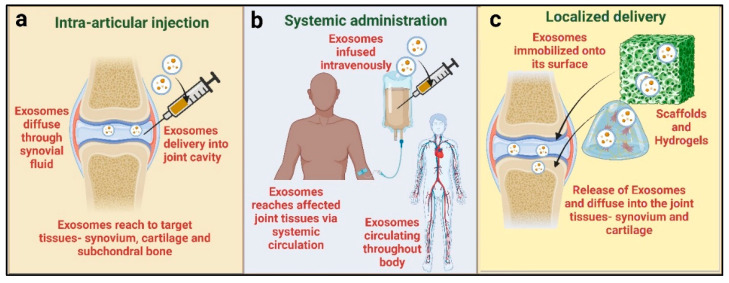
Delivery routes of exosomes for OA treatment. (**a**) Intra-articular injection: exosomes are delivered into the joint, which diffuse through the synovial fluid and reach other target tissues. (**b**) Systemic administration: exosomes are infused into the bloodstream and circulate in the whole body and cardiovascular system, reaching the affected joint tissues. (**c**) Localized delivery: exosomes are incorporated onto the scaffold structure or hydrogel, thus releasing the exosomes, which diffuse into the joint tissues for therapeutic effects. The figure was created with Bio Render (Toronto, ON, Canada).

**Figure 4 ijms-25-09149-f004:**
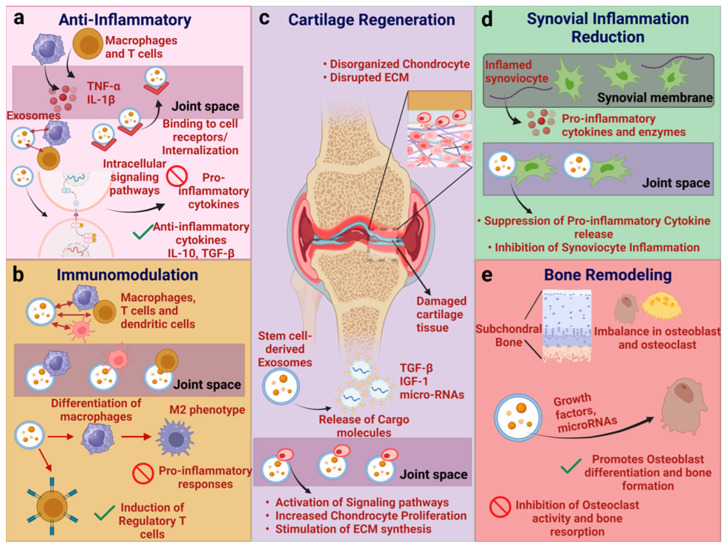
Mechanism of action by which MSC-Exos exert therapeutic responses. (**a**) Immune cells release cytokines in the joint and exosomes interact with immune cells by binding to cell surface receptors or through the internalization process. The exosomes also influence intracellular signaling pathways for the inhibition of pro-inflammatory cytokines and the promotion of anti-inflammatory cytokines. (**b**) Immune cells interact with exosomes within the joint space and promote the differentiation of macrophages towards the M2 phenotype, thereby regulating T cell activity. (**c**) Exosomes release cargo molecules involved in cartilage repair and regeneration. The internalization of exosomes leads to the activation of signaling pathways, increasing the proliferation and synthesis of the extracellular matrix, such as collagen and proteoglycans. (**d**) Exosomes interact with inflamed synoviocytes, reduce the secretion of pro-inflammatory cytokines, and decrease inflammation. (**e**) Exosomes promote osteoblast differentiation and bone formation through growth factors or microRNAs, thereby inhibiting osteoclast activity and bone resorption to regain bone integrity. The figure was created with Bio Render (Toronto, ON, Canada).

**Table 1 ijms-25-09149-t001:** List of various sources of MSC-Exos.

Type of MSC-Derived Exosomes	Source of Exosomes	Mechanism of Action	References
Adipose mesenchymal stem cell (AD-MSC)-derived exosomes	Adipose tissue (subcutaneous fat mostly from the abdomen)	Prevent ECM degradation and the reduction in inflammatory mediators	[132]
Synovial mesenchymal stem cell (SMSC)-derived exosomes	Synovial fluid	Inhibit chondrocyte apoptosis and ECM degradation	[133]
Bone marrow mesenchymal stem cell (BMSC)-derived exosomes	Bone marrow	Support the restoration of cartilage, and inhibit apoptosis and cartilage degradation	[134]
Embryonic mesenchymal stem cell (E-MSC)-derived exosomes	Embryo (inner mass of the blastocyst)	Balance cartilage extracellular matrix synthesis and degradation	[135]
Umbilical cord mesenchymal stem cell (hUCMSC)-derived exosomes	Wharton’s jelly of the umbilical cord	Promote anti-inflammation, the proliferation of chondrocytes, and immunomodulation	[136]
Dental pulp mesenchymal stem cell (DP-MSC)-derived exosomes	Dental pulp of the teeth	Inhibition of cell apoptosis, increase in matrix synthesis, and decrease in catabolic factor expression	[137]

**Table 3 ijms-25-09149-t003:** Clinical trials of stem cell-derived exosomes in OA patients.

Stem Cell-Derived Exosomes	Gender	Age Group	Clinical Phase Trial	Participants Enrolled	Country	Dosage of Exosomes	Site of Injection	Completion Status	Results	Clinical Trial ID Number
MSCs	All	30–70 years	Phase I	10	Chile	3–5 × 10^11^ cells	Intra-articular	Not completed	Not disclosed	NCT05060107
BM-MSCs, UC-MSCs, and AD-MSCs	All	40–70 years	Phase III	475	United States	2 × 10^7^ cells	Intra-articular	Completed	No adverse events and no significant change in the magnetic resonance imaging OA score compared to the baseline	NCT03818737
UC-MSCs	All	30–75 years	Phase I	24	Chile	2 × 10^6^ cells for the low dose, 20 × 10^6^ cells for the medium dose, and 80 × 10^6^ cells for the high dose	Intra-articular	Completed	Decreased inflammation and degenerative response and significant pain improvement; all of the doses were safe and no severe adverse events were reported; improvements were high in the medium- and low-dose treatments	NCT03810521
BM-MSCs	All	48–66 years	Phase II	30	Spain	40 × 10^6^ cells	Intra-articular	Completed	T2 relaxation measurements showed a decrease in poor cartilage areas and pain relief; feasible and safe	NCT01586312
BM-MSCs	All	18–76 years	Phase I/II	12	Spain	2 × 10^7^ cells	Intra-articular	Completed	T2 relaxation measurement showed a significant decrease in poor cartilage areas, the improvement of algo-functional indices from 65% to 78% by 1 year, and pain relief; feasible and safe	NCT01183728
BM-MSCs	All	40–80 years	Phase I/II	38	Spain	100 × 10^6^ cells	Intra-articular	Completed	Clinical improvement at the end of the follow-up; viable therapeutic option	NCT02365142
AD-MSCs	All	40–75 years	Phase II	106	China	Not mentioned	Intra-articular	Completed	Not disclosed	NCT04208646
AD-MSCs	All	45–65 years	Phase II	18	Saudi Arabia	1 × 10^8^ cells	Intra-articular	Completed	Not disclosed	NCT03308006
AD-MSCs	All	42–75 years	Phase I	10	Jordan	5 × 10^7^ cells	Intra-articular	Not completed	Not disclosed	NCT02966951
UC-MSCs (Wharton Jelly-derived)	All	42–75 years	Phase I	10	Jordan	5 × 10^7^ cells	Intra-articular	Not completed	Not disclosed	NCT02963727
AD-MSCs	All	18–70 years	Phase I	18	China	1 × 10^7^ cells for the low dose, 2 × 10^7^ cells for the medium dose, and 5 × 10^7^ cells for the high dose	Intra-articular	Completed	Reduction in WOMAC and SF-36 scores; multi-compositional MRI is an effective tool for evaluating cartilage repair	NCT02641860
BM-MSCs	All	18–65 years	Phase I	6	Iran	Not mentioned	Intra-articular	Completed	Not disclosed	NCT01436058
BM-MSCs	All	18–65 years	Phase II	40	Iran	Not mentioned	Intra-articular	Completed	Not disclosed	NCT01504464
AD-MSCs	All	40–70 years	Phase I/II	18	China	1 × 10^7^ cells for the low dose, 2 × 10^7^ cells for the medium dose, 5 × 10^7^ cells for the high dose	Intra-articular	Completed	Safe, alleviates pain, improves cartilage volume and function; 5 × 10^7^ cells exhibited the highest improvement	NCT01809769
BM-MSCs	All	30–70 years	Phase I/II	10	India	Not mentioned	Intra-articular	Not completed	Not disclosed	NCT01152125
UC-MSCs	All	Up to 70 years	Phase II	60	China	1 × 10^7^ cells for the low dose and 2 × 10^7^ cells for the high dose	Intra-articular	Not completed	Not disclosed	NCT03383081
AD-MSCs	All	50–70 years	Phase I	4	Taiwan	8–10 × 10^6^ cells	Intra-articular	Completed	Not disclosed	NCT02544802
BM-MSCs	All	40–75 years	Phase I/II	24	India	10 × 10^6^ cells	Intra-articular	Not completed	Not disclosed	NCT01985633
DP-MSCs	All	40–70 years	Phase I	60	China	Not mentioned	Intra-articular	Not completed	Not disclosed	NCT04130100
BM-MSCs	All	18–70 years	Phase II	50	Malaysia	Not mentioned	Intra-articular	Not completed	Not disclosed	NCT01459640

## Data Availability

Not applicable.

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
