# Peer review of "Mesenchymal Stem Cell-Derived Exosomes as a Treatment Option for Osteoarthritis"

_ijms, 2024, doi:10.3390/ijms25179149_

Round 1

Reviewer 1 Report

Comments and Suggestions for Authors

The review "Mesenchymal Stem Cells-Derived Exosomes as Treatment Option for Osteoarthritis", addresses the important topic of the possibility of using exosomes derived from MSCs in OA therapy.

The work is prepared correctly, but I have minor comments:

- the aim of the work should be clearly stated at the end of the introduction

- I propose to supplement the review with issues concerning the limitations of the use of exosomes in the scope of the problem of therapeutic quality of exosomes; quality/quantity of exosome cargo; repeatability or limitations regarding MSC donors.

Author Response

  • the aim of the work should be clearly stated at the end of the introduction

   Answer: Thank you for your concern. We have explained the aims of this study at the end of the   introduction which explains it precisely the need of the MSC-Exos therapy for the treatment of OA which could have anti-inflammatory effects, can modulate the immune response, cartilage regeneration, decrease synovial inflammation and helpful in bone remodeling. Maybe in the future, this therapy could have fruitful impact on the OA patients but still it needs to be explored more.

  • I propose to supplement the review with issues concerning the limitations of the use of exosomes in the scope of the problem of therapeutic quality of exosomes; quality/quantity of exosome cargo; repeatability or limitations regarding MSC donors.

Answer: Thank you for underlying the important point of the clinical translation. We have explained the limitations of exosomes in all the aspects in section 7.

Reviewer 2 Report

Comments and Suggestions for Authors

Please avoid to repeat the same information more than once. Authors should check the whole manuscript and better organized the concept trying to avoid repetitions.

Lines 39-40: reference 3 is not appropriate as it is a study on adenosine receptors in OA.

Lines 42-43: this part should be improved. OA is a whole joint disease involving all joint tissues and not only cartilage damaged. OA is characterized by subchondral bone remodelling, meniscal degeneration, inflammation and fibrosis of both infrapatellar fat pad and synovial membrane (which function as an anatomo-funtional unit).

Reference 5 is not focused on cartilage damage in OA.

Lines 43-46: authors reported that the knee is the most affected joint also at line 38. In any case reference 6 is not appropriate as it does not support the claim stated by the authors as it is a study on the potential of stem cells therapy for OA.

Authors should check all the references to ensure that they are appropriate.

Line 48: again, reference 8 is not on cartilage but on the advantages and challenges of stem cell therapy for OA.

Lines 48-51: this part should be moved to lines 40-42 in order to avoid repeating the same concepts.

Lines 51-55 and lines 116-127: authors should add that there is not only a change in the chondrocyte biomechanical changes but also biomechanical chondrocytes (DOI:10.3390/biomedicines11071942 etc).

Lines 109-112: authors forget meniscal degeneration and inflammation and fibrosis of the infrapatellar fat pad. Reference 35 is a review only on chondrocytes.

Lines 127-129: synovial inflammation may be triggered also by DAMPs released by cartilage breakdown.

Line 130: osteoarthritis should be OA.

Abbreviations should be defined at first mention and only once. For example, MMP-1 etc at line 125, IL.1beta etc at line 131 and so on.

Section 2 should be implemented as it is focused only on cartilage and OA and a few sentences on synovial membrane. What about other tissues involved? Is there a reason why authors focused only on cartilage?

Section 3. Authors should check and refer to EULAR and ACR guidelines for reporting the OA treatments.

Lines 181-182: references should be added.

Lines 187-189: this is the first time that authors mentioned “reliable biomarkers” in the text. What are these reliable biomarkers? What is the link between these biomarkers and the need of studies with long-term follow-up?

Line 197: “Autologous mesenchymal stem cells (MSCs)” derived from what tissue?

Line 198: from what adipose tissue? autologous ADSCs?

Figure 1 needs to be improved. First, in the legend it is written “comparison between MSC-exos …” but in the figure there is no MSC-exos. It should be checked. Moreover, authors claimed that MSCs have an higher efficacy compared to whole stem cell therapy and traditional pharmaceutics. Is there one or more meta-analysis supporting this? Claims should be supported by appropriate references.

Figures were created with Biorender. Did the authors submit the licenses?

Line 248: EV were defined earlier. Authors should be check.

Lines 248-250: this part is already reported prevouoily in the manuscript.

Line 251: exosomes were defined in the previous section. Moreover, here the dimensions are (50-150 nm in diameter), while 30-150 nm is reported at line 210.

Lines 262-263: what is the link between vesicular apoptosis and exosomes/macrovesicles?

Lines 290-292:please delete this part which has been repeated several times.

Lines 316-320: authors should clarify if this method is currently used and for what diseases.

Lines 320-326: again is this method used?

Line 332: here it is reported that the 3 methods are currently used for the OA treatment in clinical practice. This point needs to be explained. References that support this must be added.

Section 5: authors focused on the effects of MSC-exo on chondrocytes. What about the effects of MSC-exo on the other tissues of the joints? Lines 363-367: this part should be expanded and references should be provided.

Lines 360-363: this part should be deleted as it is a repetition.

Title of section 5.1.1. is unclear.

Line 399: what MSCs?

Line 403: “al-pha” should be corrected.

Section 5.1.2: could the author specify adipose tissue type?

Lines 536-540: exosomes isolated from OA infrapatellar fat pad?  Caution should be paid when using infrapatellar fat pad stem cells as it has been shown that OA infrapatellar fat pad MSCs seem to be primed by the inflammatory environment.

Table 1: Could the authors add also studies mentioned in section 5?

Lines 625-627: are there systematic reviews with meta-analysis supporting this claiming?

Line 649: what adipose tissue?

Table 2 should be improved. Dose of exosomes, site of injection (intraarticular?)  and results should be added.

Comments on the Quality of English Language

Minor revision is needed.

Author Response

  • Please avoid to repeat the same information more than once. Authors should check the whole manuscript and better organized the concept trying to avoid repetitions.

Answer: Thank you for your kind suggestion. We have revised the manuscript and have removed repeated parts.

  • Lines 39-40: reference 3 is not appropriate as it is a study on adenosine receptors in OA.

Answer: Thank you for the suggestion. We have updated the reference, Lines 39-40.

  • Lines 42-43: this part should be improved. OA is a whole joint disease involving all joint tissues and not only cartilage damaged. OA is characterized by subchondral bone remodelling, meniscal degeneration, inflammation and fibrosis of both infrapatellar fat pad and synovial membrane (which function as an anatomo-funtional unit).

Answer: Thank you very much for the valuable suggestion. We have added the information, Lines 46-49.

  • Reference 5 is not focused on cartilage damage in OA.

Answer: Thank you for your comment. We have made corrections in reference.

Reference: Sowers, M. Epidemiology of risk factors for osteoarthritis: systemic factors. Current Opinion in Rheumatology 2001, 13, 447-451. doi: 10.1097/00002281-200109000-00018.

  • Lines 43-46: authors reported that the knee is the most affected joint also at line 38. In any case reference 6 is not appropriate as it does not support the claim stated by the authors as it is a study on the potential of stem cells therapy for OA.

Answer: Thank you for your valuable suggestion. We have updated the reference as follows:

Reference: Loeser, R.F.; Goldring, S.R.; Scanzello, C.R.; Goldring, M.B. Osteoarthritis: a disease of the joint as an organ. Arthritis and rheumatism 2012, 64, 1697. doi: 10.1002/art.34453

  • Authors should check all the references to ensure that they are appropriate.

Answer: Thank you for the suggestions. We have rechecked the references and updated wherever it was required.

  • Line 48: again, reference 8 is not on cartilage but on the advantages and challenges of stem cell therapy for OA.

            Answer: Thank you for your suggestion. We have updated the reference as follows:

Reference: Ng, H.Y.; Lee, K.-X.A.; Shen, Y.-F. Articular cartilage: structure,   composition, injuries and repair. JSM Bone Joint Dis 2017, 1, 1010.

  • Lines 48-51: this part should be moved to lines 40-42 in order to avoid repeating the same concepts.

Answer: Thank you for the suggestion. We have moved the suggested part to Lines 40-44.

  • Lines 51-55 and lines 116-127: authors should add that there is not only a change in the chondrocyte biomechanical changes but also biomechanical chondrocytes (DOI:10.3390/biomedicines11071942 etc).

Answer: Thank you for the suggestions. We have added the information, Lines 64-72.

Reference: Pettenuzzo, S.; Arduino, A.; Belluzzi, E.; Pozzuoli, A.; Fontanella, C.G.; Ruggieri, P.; Salomoni, V.; Majorana, C.; Berardo, A. Biomechanics of Chondrocytes and Chondrons in Healthy Conditions and Osteoarthritis: A Review of the Mechanical Characterisations at the Microscale. Biomedicines 2023, 11, doi:10.3390/biomedicines11071942.

  • Lines 109-112: authors forget meniscal degeneration and inflammation and fibrosis of the infrapatellar fat pad. Reference 35 is a review only on chondrocytes.

Answer: Thank you for the suggestion. We have merged this part in introduction section and have added relevant references, Lines 46-50.

  • Lines 127-129: synovial inflammation may be triggered also by DAMPs released by cartilage breakdown.

Answer: Thank you for the suggestion. We have merged this part in introduction section and have added relevant references, Lines 81-83.

  • Line 130: osteoarthritis should be OA.

Answer: Thank you for the reminding us. We have made correction, Line 85.

  • Abbreviations should be defined at first mention and only once. For example, MMP-1 etc at line 125, IL.1beta etc at line 131 and so on.

Answer: Thank you for the suggestions. We have made corrections, Lines 86-87 and 89.

  • Section 2 should be implemented as it is focused only on cartilage and OA and a few sentences on synovial membrane. What about other tissues involved? Is there a reason why authors focused only on cartilage?

Answer: Thank you for the valuable question. We checked the literature and found that most of the studies were on cartilage and only a few were about the other tissues. That is why most of the references are related to cartilage and OA.

  • Section 3. Authors should check and refer to EULAR and ACR guidelines for reporting the OA treatments.

Answer: Thank you very much for your valuable suggestion. We have checked the guidelines and added the relevant references, Lines 173-180.

References:  

  1. Moseng, T.; Vlieland, T.P.V.; Battista, S.; Beckwée, D.; Boyadzhieva, V.; Conaghan, P.G.; Costa, D.; Doherty, M.; Finney, A.G.; Georgiev, T. EULAR recommendations for the non-pharmacological core management of hip and knee osteoarthritis: 2023 update. Annals of the rheumatic diseases 2024, 83, 730-740. doi: 10.1136/ard-2023-225041
  2. Kolasinski, S.L.; Neogi, T.; Hochberg, M.C.; Oatis, C.; Guyatt, G.; Block, J.; Callahan, L.; Copenhaver, C.; Dodge, C.; Felson, D. 2019 American College of Rheumatology/Arthritis Foundation guideline for the management of osteoarthritis of the hand, hip, and knee. Arthritis & rheumatology 2020, 72, 220-233. doi:10.1002/art.41142.

  • Lines 181-182: references should be added.

Answer: Thank you for the comment. We have added the reference as follows:

Reference: Li, S.; Cao, P.; Chen, T.; Ding, C. Latest insights in disease-modifying osteoarthritis drugs development. Ther Adv Musculoskelet Dis 2023, 15, 1759720x231169839, doi:10.1177/1759720x231169839.

  • Lines 187-189: this is the first time that authors mentioned “reliable biomarkers” in the text. What are these reliable biomarkers? What is the link between these biomarkers and the need of studies with long-term follow-up?

Answer: Thank you for the question. Reliable markers here are the biological indicators used clinically to diagnose or monitor osteoarthritis. Aggrecan fragments, hyaluronic acid level, cytokines like IL-6, IL-1, etc. are some biomarkers used in OA diagnosis. Long term follow up of these biomarkers is required to track the changes over time of disease, to understand the relationship between biomarkers and clinical outcome, and to evaluate the treatment efficacy.

  • Line 197: “Autologous mesenchymal stem cells (MSCs)” derived from what tissue?

Answer: Thank you for the question. Autologous mesenchymal stem cells (MSCs) are mostly derived from bone and adipose tissue.

Reference: Pittenger, M. F., Discher, D. E., Péault, B. M., Phinney, D. G., Hare, J. M., & Caplan, A. I. (2019). Mesenchymal stem cell perspective: cell biology to clinical progress. NPJ Regenerative medicine4(1), 22. https://doi.org/10.1038/s41536-019-0083-6

  • Line 198: from what adipose tissue? autologous ADSCs?

Answer: Thank you for the question. Autologous ADSCs mostly derived from abdominal subcutaneous fats.

Reference: Lee, S., Chae, D. S., Song, B. W., Lim, S., Kim, S. W., Kim, I. K., & Hwang, K. C. (2021). ADSC-based cell therapies for musculoskeletal disorders: a review of recent clinical trials. International Journal of Molecular Sciences22(19), 10586.  https://doi.org/10.3390/ijms221910586

  • Figure 1 needs to be improved. First, in the legend it is written “comparison between MSC-exos …” but in the figure there is no MSC-exos. It should be checked. Moreover, authors claimed that MSCs have an higher efficacy compared to whole stem cell therapy and traditional pharmaceutics. Is there one or more meta-analysis supporting this? Claims should be supported by appropriate references.

Answer: Thank you for your valuable opinion. We have made changes in the figure as we would like to draw attention towards the MSC derived exosomes which is much better than the other treatments. Actually, there are no studies comparing all the three therapies together but we have checked the other studies stating that they exhibit higher therapeutic efficacy and much safer for use, as we are already aware of the NSAIDs which are effective though but causes severe GI, renal and cardiovascular toxicity. So, we compiled the data from different studies and according to our understanding we illustrated in the form of figure. Some of the references which are best suited for this statement are:

-Lamo-Espinosa, J.M., Blanco, J.F., Sánchez, M. et al. Phase II multicenter randomized controlled clinical trial on the efficacy of intra-articular injection of autologous bone marrow mesenchymal stem cells with platelet rich plasma for the treatment of knee osteoarthritis. J Transl Med 18, 356 (2020). https://doi.org/10.1186/s12967-020-02530-6

-Zhang, Z., Zhao, S., Sun, Z. et al. Enhancement of the therapeutic efficacy of mesenchymal stem cell-derived exosomes in osteoarthritis. Cell Mol Biol Lett 28, 75 (2023). https://doi.org/10.1186/s11658-023-00485-2

-Ragni, Enrico, Carlotta Perucca Orfei, Paola De Luca, Alessandra Colombini, Marco Viganò, Gaia Lugano, Valentina Bollati, and Laura de Girolamo. "Identification of miRNA reference genes in extracellular vesicles from adipose derived mesenchymal stem cells for studying osteoarthritis." International Journal of Molecular Sciences 20, no. 5 (2019): 1108. https://doi.org/10.3390/ijms20051108

  • Figures were created with Biorender. Did the authors submit the licenses?

Answer: Thank you for your kind concern. Currently we used a trial version of Bio- Render. We will apply for a licensee for further use of the software.

  • Line 248: EV were defined earlier. Authors should be check.

Answer: Thank you for the valuable suggestion. We have deleted the repeated information.

  • Lines 248-250: this part is already reported previously in the manuscript.

Answer: Thank you for the comment. We have removed the repeated information.

  • Line 251: exosomes were defined in the previous section. Moreover, here the dimensions are (50-150 nm in diameter), while 30-150 nm is reported at line 210.

Answer: Thank you for the valuable comment. We have made the correction in exosome size information.

  • Lines 262-263: what is the link between vesicular apoptosis and exosomes/macrovesicles?

Answer:  Thank you for the question. They both are types of extracellular vesicles of different sizes.

  • Lines 290-292: please delete this part which has been repeated several times.

Answer: Thank you very much for your kind suggestion. We have removed the repeated information.

  • Lines 316-320: authors should clarify if this method is currently used and for what diseases.

Reference: Thank you for the suggestions. We have added the information, in Line 367.

  • Lines 320-326: again is this method used?

Answer: Thank you for the suggestion. We have added the reference regarding the method, Lines 370-372.

Reference:  Sang, X.; Zhao, X.; Yan, L.; Jin, X.; Wang, X.; Wang, J.; Yin, Z.; Zhang, Y.; Meng, Z. Thermosensitive hydrogel loaded with primary chondrocyte-derived exosomes promotes cartilage repair by regulating macrophage polarization in osteoarthritis. Tissue Engineering and Regenerative Medicine 2022, 19, 629-642.  doi: 10.1007/s13770-022-00437-5

  • Line 332: here it is reported that the 3 methods are currently used for the OA treatment in clinical practice. This point needs to be explained. References that support this must be added.

Answer: Thank you for your valuable concern. We would like to address that we have stated the above studies with the involvement of these methods in practice from Line 311-331 with their respective references.

  • Section 5: authors focused on the effects of MSC-exo on chondrocytes. What about the effects of MSC-exo on the other tissues of the joints? Lines 363-367: this part should be expanded and references should be provided.

Answer: Thank you for your valuable concern. Chondrocytes are one of the important components of articular cartilage and their dysregulation is a major issue in OA. Therefore, most of the studies focused on chondrogenesis in OA. That is why we referred to chondrocytes. The effect of MSC-Exos on other tissue has been discussed subsections. For the part Lines 363- 367, this is the explanation of the figure we have created from the scratch and is not been published in form of figure anywhere which is why we could not provide the references for the figures we created.

  • Lines 360-363: this part should be deleted as it is a repetition.

Answer: Thank you for comment. We have removed the repeated part.

  • Title of section 5.1.1. is unclear.

Answer: Thank you for your valuable question. We have rechecked the information regarding the source of MSC-Exos in section 4.1.1, and merged the information in the related section 4.1.2. Exosomes in all these studies were derived from bone marrow mesenchymal stem cells.

  • Line 399: what MSCs?

Answer: Thank you for your concern. We have rechecked the information regarding the source of MSC-Exos in section 4.1.1 and merged the information in the related section 4.1.2. Exosomes in all these studies were derived from bone marrow mesenchymal stem cells.

  • Line 403: “al-pha” should be corrected.

Answer: Thank you for the comment. We have corrected, Line 550.

  • Section 5.1.2: could the author specify adipose tissue type?

Answer:  Thank you for your question. We have added the information, Lines 442-443, 445-446, and 449-450.

  • Lines 536-540: exosomes isolated from OA infrapatellar fat pad?  Caution should be paid when using infrapatellar fat pad stem cells as it has been shown that OA infrapatellar fat pad MSCs seem to be primed by the inflammatory environment.

Answer: Thank you for your comment. We checked the information regarding this issue. In a study it was reported that infrapatellar fat pad-mesenchymal stem cells from both healthy or knee OA were suitable for therapeutic application. In that study, they found that an Inflammatory OA environment does not prime IFP-MSCs to pro-inflammatory properties.

Reference: Chen, H. H., Chen, Y. C., Yu, S. N., Lai, W. L., Shen, Y. S., Shen, P. C., Lin S.H, Chang C.H. & Lee, S. M. (2022). Infrapatellar fat pad-derived mesenchymal stromal cell product for treatment of knee osteoarthritis: a first-in-human study with evaluation of the potency marker. Cytotherapy24(1), 72-85. DOI: 10.1016/j.jcyt.2021.08.006

  • Table 1: Could the authors add also studies mentioned in section 5?

Answer: Thank you for your suggestion. As it is already mentioned in the main text above, it would be just repetition of the information so therefore, we would like to keep it precise and content.

  • Lines 625-627: are there systematic reviews with meta-analysis supporting this claiming?

Answer: Thank you for concern. Yes, these are systematic studies with clinical data confirming their claim in terms of safety and efficacy for the OA treatment.

-McIntyre JA, Jones IA, Han B, Vangsness CT Jr. Intra-articular Mesenchymal Stem Cell Therapy for the Human Joint: A Systematic Review. Am J Sports Med. 2018 Dec;46(14):3550-3563. doi: 10.1177/0363546517735844. Epub 2017 Nov 3. PMID: 29099618.

-Yubo M, Yanyan L, Li L, Tao S, Bo L, Lin C (2017) Clinical efficacy and safety of mesenchymal stem cell transplantation for osteoarthritis treatment: A meta-analysis. PLoS ONE 12(4): e0175449. https://doi.org/10.1371/journal.pone.0175449

  • Line 649: what adipose tissue?

Answer: Thank you for your valuable insight. The adipose-derived MSCs were prepared from the abdominal subcutaneous fats. We have also added the information in the main text.

  • Table 2 should be improved. Dose of exosomes, site of injection (intraarticular?)  , and results should be added.

Answer: Thank you for your valuable insight. We have included the dosage, injection site, and results in the table.

Reviewer 3 Report

Comments and Suggestions for Authors

The manuscript by Vadhan et al. reviews the potential of mesenchymal stem cell-derived exosomes (MSC-Exos) as a treatment strategy for osteoarthritis (OA). This topic is particularly relevant because OA is a major cause of pain and disability among the elderly worldwide, and currently, there are no effective treatments for the disease. The manuscript is well written and offers a detailed discussion on the topic, highlighting the promise of MSC-Exos as an alternative method for treating and alleviating the symptoms of OA, such as bone/cartilage degradation and inflammation reduction. The authors also briefly discuss the limitations of using MSC-Exos for treating OA.

In my opinion, the manuscript can be accepted for publication after addressing the following minor issues:

  1. Please combine Sections 1 and 2 and transfer some information about MSC-Exos to Section 3 to provide a clearer picture of OA and the pathophysiology of the disease, emphasizing the need to develop more effective treatments. Subsequently, provide a concise description of the potential of MSC-Exos. In the revised introduction's final part, discuss the aims of your work.
  2. Please provide the full names of all abbreviations when they first appear in the text (e.g., TNF-α).
  3. Section 3.1 should be more informative, as non-pharmacological approaches are typically combined with pharmacological ones. Also, in the heading of Section 3.1, please change "approach" to "approaches."
  4. Figure 1: The font size in the panels describing efficiency is too small.
  5. In Section 5.1, I suggest the authors provide a table to summarize the main types of MSC-Exos.
  6.  Please elaborate on the limitations of using MSC-Exos as a strategy for treating OA

Author Response

  • Please combine Sections 1 and 2 and transfer some information about MSC-Exos to Section 3 to provide a clearer picture of OA and the pathophysiology of the disease, emphasizing the need to develop more effective treatments. Subsequently, provide a concise description of the potential of MSC-Exos. In the revised introduction's final part, discuss the aims of your work.

Answer: Thank you for the valuable suggestions. We have merged sections 1 and 2 and have added further information in section 3, marked with yellow color, lines 59-72 and 259-267. Also, aims have been added in the introduction’s final part, lines 131-141.

  • Please provide the full names of all abbreviations when they first appear in the text (e.g., TNF-α).

Answer: Thank you for your kind suggestion. We have added the information, Lines, 86-87.

  • Section 3.1 should be more informative, as non-pharmacological approaches are typically combined with pharmacological ones. Also, in the heading of Section 3.1, please change "approach" to "approaches."

Answer: Thank you for your kind suggestion. We have added details in this section. Also, heading of Section 3.1(now is 2.1) has been corrected.

  • Figure 1: The font size in the panels describing efficiency is too small.

Answer: Thank you for your concern. We have made the necessary changes in the figure.

  • In Section 5.1, I suggest the authors provide a table to summarize the main types of MSC-Exos.

Answer: Thank you for your valuable suggestion. We have added Table 1 to summarize the main types of MSC-Exos.

  •  Please elaborate on the limitations of using MSC-Exos as a strategy for treating OA

Answer: Thank you for the valuable comment. We have added further information in section 7, marked with yellow color.

Round 2

Reviewer 2 Report

Comments and Suggestions for Authors

Each figure created with Biorender needs a licence to be published. 

Author Response

Each figure created with Biorender needs a licence to be published.

Answer: Thank you very much for the valuable suggestion. We have applied for the BioRender license, under the account information -  [email protected] and Invoice number 23CF7476-0001.